# Efficient Distortion Mitigation and Partition Reduction in Mapping Global Geodata: Dual Orthogonal Equidistant Cylindrical Projection Approach

Aleksandar Dimitrijević *, Aleksandar Milosavljević and Dejan Rančić

Faculty of Electronic Engineering, University of Niš, 18104 Niš, Serbia;
aleksandar.milosavljevic@elfak.ni.ac.rs (A.M.); dejan.rancic@elfak.ni.ac.rs (D.R.)
* Correspondence: aleksandar.dimitrijevic@elfak.ni.ac.rs; Tel.: +381-69-10-456-00

**Abstract:** The rapid growth in Earth's global geospatial data necessitates an efficient system for organizing the data, facilitating data fusion from diverse sources, and promoting interoperability. Mapping the spheroidal surface of the planet presents significant challenges as it involves balancing distortion and splitting the surface into multiple partitions. The distortion decreases as the number of partitions increases, but, at the same time, the complexity of data processing increases since each partition represents a separate dataset and is defined in its own local coordinate system. In this paper, we propose the Dual Orthogonal Equidistant Cylindrical projection method to mitigate distortion and reduce the number of partitions. Additionally, we use the rotation of the graticule system on the globe to achieve the oblique aspect, which effectively minimizes average angular and areal distortions of Earth's landmass and reduces the interruption of continental plates caused by partition edges. By incorporating auxiliary latitudes and proposing an approximate authalic latitude, we further enhance the mapping of the ellipsoid onto the sphere, simplifying calculations. The experimental results demonstrate a substantial reduction in distortion and interruption of continental plates. With only two partitions, an average landmass angular distortion of less than 3.56 degrees and an average areal distortion of less than 1.07 were achieved.

**Keywords:** discrete global grid system; equidistant cylindrical projection; Yin–Yang; distortion

## 1. Introduction

Organizing and referencing geospatial data pose increasingly complex challenges due to the sheer volume and rapid growth of the data collected. For example, the daily influx of high-resolution satellite imagery alone amounts to terabytes of data. It is important to store these data in a format that allows easy access, referencing, sharing, and analysis without frequent re-projections to maintain accuracy.

Addressing these challenges requires the development of a spatial reference frame capable of fusing data from diverse sources into a global mosaic at multiple resolutions. Discrete Global Grid Systems (DGGSs) have emerged as a promising class of such reference frames. They use hierarchical tessellation to partition and address the entire planet without gaps or overlaps. The development of DGGS began in the mid 20th century [1] but only became popular at the turn of the century [2–4]. A prominent subclass of DGGSs are Geodesic DGGSs (GDGGSs) [4], which project the surface of the planet onto the faces of regular or semi-regular circumscribed polyhedra. Commonly used base polyhedra are the five Platonic solids, in particular the hexahedron (cube), while the most common semi-regular polyhedron is the truncated icosahedron. The appearance of unfolded regular polyhedra, the number and shape of faces, and the way they partition Earth's surface can be observed in Figure 1.

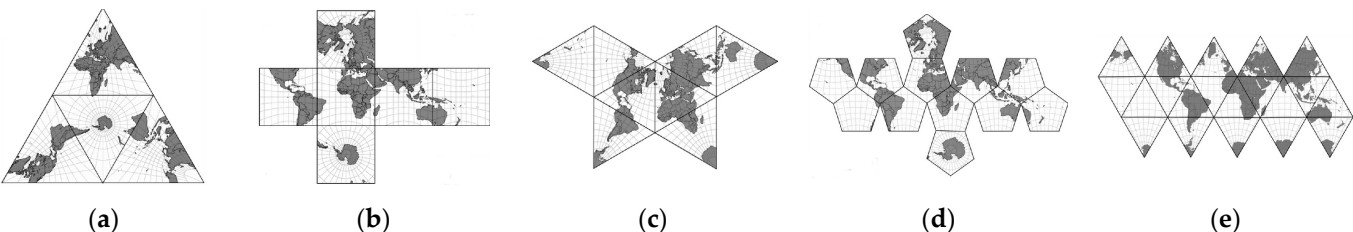

**Figure 1.** Unfolded regular polyhedra used in GDDGSs: (**a**) tetrahedron; (**b**) hexahedron; (**c**) octahedron; (**d**) dodecahedron; (**e**) icosahedron.

While increasing the number of faces in a polyhedron improves its approximation of a spherical surface, it introduces complications when merging adjacent partitions due to separate coordinate systems and datasets. Moreover, memory and CPU consumption increase with the number of partitions in systems handling streaming geospatial data, like out-of-core planet-scale terrain rendering applications [5]. Therefore, minimizing the number of partitions becomes desirable. This paper explores the use of Dual Orthogonal Equidistant Cylindrical projection, also known as the Yin–Yang grid [6], to reduce the number of partitions to two. While the Yin–Yang grid has found applications in various fields, its potential as a cartographic projection has not been sufficiently explored. Additionally, this paper investigates the rotation of the graticule system on the globe to achieve the oblique aspect, which reduces the distortion of Earth's landmass and minimizes disruption of continental plates caused by partition boundaries. The utilization of auxiliary latitudes in reducing distortion during the mapping of the ellipsoid onto the sphere is also considered.

This paper is divided into five sections. After this brief introduction, Section 2 provides an overview of the historical development and importance of organizing global geospatial data, as well as the emergence and standardization of DGGSs. Section 3 addresses the projection of geospatial data onto a plane using the proposed method, which encompasses ellipsoid to sphere mapping using auxiliary latitudes, sphere to plane mapping using Dual Orthogonal Equidistant Cylindrical projection, and the rotation of the graticule system. The goal is to minimize distortions and the number of partitions. The experimental results and discussion are presented in Section 4, followed by the conclusion in Section 5.

## 2. Related Work

The need to develop a system for organizing global geospatial data is not new. One of the first studies on the feasibility of implementing the Earth Data Base System [7] was conducted in the early 1970s for the needs of the US Navy. The system was based on Quadrilateralized Spherical Cube (QSC), one of the first hexahedral projections implemented on digital computers. The proposed system was soon modified [8], and, in the following years, it was also used as part of the Cosmic Background Explorer (COBE) project at NASA.

Due to the regular and uniform structure of the grid consisting of square cells, consistency with the Cartesian coordinate system, ease of interpolation and extrapolation, and straightforward visualization, hexahedral projections have gained wide popularity and are used in many different fields.

Hierarchical Equal-Area isoLatitude Pixelization (HEALPix) is a class of spherical projections with the property of distributing $12N^2$ points as uniformly as possible over the surface of the unit sphere [9]. These hybrid projections combine the Lambert cylindrical equal-area projection for the equatorial region with the interrupted Collignon projection for the polar regions. Of this infinite class of projections, only the projection with three base resolution pixel layers between the north and south poles and four equatorial base resolution pixels can be rearranged to a hexahedral projection.

Rotated HEALPix (rHEALPix) [10] is an extension of the HEALPix scheme that introduces rotation capabilities, is better adapted to standards, and inherently combines polar triangles into quadratic partitions. Further, rHEALPix has found wide application in organizing global geospatial data [10–12]. All the previously mentioned projections

are equal-area, but with significant angular distortions and even discontinuities. Due to their simpler implementation and relatively good balance between angular and areal distortion, many hexahedral projections also find application in computer graphics [13] and, in particular, in planet-scale terrain visualization [14]. Among the best known are Adjusted Spherical Cube (ASC) [15], Continuous Cube Mapping (CCM) [16], and Cartesian Spherical Cube (CSC) [17].

In addition to polyhedral projections, the UTM and Equi7 grid systems and equidistant cylindrical projection, also known as plate carree, are commonly used today. Equidistant cylindrical projection, despite its considerable distortion, is still widely used to organize global data because it is simple, has a regular grid structure, and can cover the entire world in a single rectangular map. The Universal Transverse Mercator (UTM) grid system, on the other hand, divides Earth into 60 zones, which reduces distortion but limits its application to medium-scale mapping. Equi7 [18] is an example of a compromise that combines seven continental grids based on equidistant azimuthal projections.

The expansion of the system for organizing and referencing global geospatial data occurs at the turn of the century, when the first classifications appear and the term Discrete Global Grid System (DGGS) is introduced for a very significant class of such systems. The importance of DGGSs is also reflected in the fact that the Open Geospatial Consortium (OGC) established the DGGS Standard and Domain Working Groups to support the standardization of these systems. In 2017, the OGC published the first version of the DGGS Abstract Specification [19]. The standardization process continued, resulting in a formal specification defined by ISO 19170-1:2021 standard [20] and a revised version of the OGC Abstract Specification [21]. The popularity of DGGS has also increased due to numerous open source implementations [22].

A recent trend in geospatial data processing involves the implementation of datacubes based on DGGSs [23], enabling efficient management of big data workflows. DGGSs, as a standardized representation of Earth, provide the foundational platform for Digital Earth [24]. Digital Earth is a concept that aims to create an interactive digital replica of the entire planet, fostering a shared understanding of the relationships between the physical and natural environment and society [25].

The design of GDGGSs is characterized by five fundamental elements [4]:

- A regular base polyhedron;
- The orientation of the base polyhedron with respect to the planet;
- A hierarchical spatial partitioning of the polyhedron faces;
- The mapping of a spherical or ellipsoidal surface to polyhedral faces and vice versa;
- Methods for indexing and addressing cells.

In the next section, we propose improvements to three of these properties of GDGGSs, aiming to reduce the number of partitions while minimizing distortion effects. Specifically, we replace the faces of a regular polyhedron with only two projection planes onto which Earth's surface is mapped using Dual Orthogonal Equidistant Cylindrical projection, called the Yin–Yang grid [6]. The graticule system on the globe is rotated to achieve the oblique aspect that minimizes the landmass distortion. We also consider the use of auxiliary latitudes in mapping the ellipsoid onto the sphere. Hierarchical spatial partitioning and methods for indexing and addressing cells are beyond the scope of this article.

## 3. Method Description

Organizing geospatial data of planet Earth is a significant challenge, especially given the recent influx of large volumes of data from various sensors that need to be integrated into a coherent mosaic while ensuring accessibility and interoperability. This data organization should be efficient in terms of

- Storage—using a compact distributed approach;
- Addressing and indexing—enabling easy and fast data access and supporting spatial and temporal localization;

- Analysis—ensuring data are in a suitable form for processing, preferably without the need for re-projection during use and with minimal loss of precision in transformations;
- Visualization—storing data in a format suitable for display.

Meeting all these requirements simultaneously is not easy given the diverse applications for geospatial data. Many applications focus solely on Earth's surface, with data recorded as two-dimensional raster layers, thereby determining the shape of the referencing system. However, projecting the spheroidal surface of the planet onto a plane has been a longstanding challenge for cartographers. No transformation fully preserves all the properties of surface entities, ensuring that they retain their shape, proportionality, and continuity. Conformal projections preserve shape but distort area significantly, while equal-area projections preserve area but distort shape considerably. A projection that is both conformal and equal-area does not exist. Preserving one property more effectively comes at the expense of the other. Furthermore, representing the entire planet's surface on a single plane without singularities or discontinuities is not possible. The use of a single planar projection leads to singularities, typically occurring at the poles or along the equator, resulting in extreme distortions. On the other hand, combining multiple projection planes, such as faces of a circumscribed polyhedron, leads to sudden changes or breaks in the representation of geographic features caused by the transition from one projection system to another.

The goal of the global grids is to create a uniform tessellation of the planet's surface without gaps and a unique cell-addressing system. Due to the planar organization of data, it becomes necessary to partition the surface into multiple sections. Increasing the number of partitions reduces distortion but complicates data manipulation when merging two or more partitions as each partition employs its own local coordinate system. These partitions are further divided into sections, which consist of blocks of data suitable for retrieval and processing. Subsequently, the sections are subdivided into smaller units known as cells, which represent the smallest addressable units in the system. Figure 2 illustrates the process of transforming the planet's surface into an addressable system of cells using the example of a subdivision into two partitions based on Dual Orthogonal Equidistant Cylindrical projection.

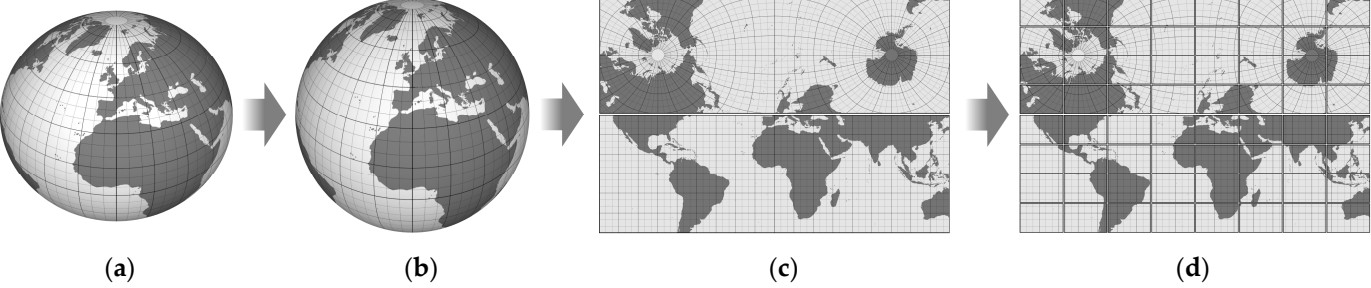

**Figure 2.** The process of transforming the surface of the planet into an addressable system of cells. The ellipsoid (**a**) is mapped onto the sphere (**b**) and the sphere is mapped onto a set of planar surfaces (**c**). A projection of a part of the planet onto a planar surface is called a partition. Partitions are further divided into smaller units called sections (**d**).

### 3.1. Mapping an Ellipsoid to a Sphere

The ellipsoid is the most commonly used approximation for the shape of the planet Earth today. Due to the widespread utilization of the Global Positioning System (GPS) and the abundance of data collected using this reference frame, the WGS 84 ellipsoid [26] serves as the primary model for creating global datasets. To ensure interoperability, the NGA (National Geospatial-Intelligence Agency) closely aligns the WGS 84 reference frame with other standards, particularly the International Terrestrial Reference Frame (ITRF) [27]. Consequently, the latest official revision of the WGS 84 reference frame (G2139) remains consistent with the IGb14 realization of the ITRF2014 [28].

In geodetic computations, the ellipsoid model is often substituted with the spherical model due to its higher symmetry and simpler calculation of many geodetic formulas. Since the ellipsoidal model deviates slightly from a perfect sphere in the case of Earth, the spherical formulas can be applied to the ellipsoid by replacing the geodetic latitude by one of the "auxiliary latitudes". Introducing a mapping from an ellipsoid to a sphere introduces an additional distortion that varies depending on the applied auxiliary latitude. Snyder mentions six auxiliary latitudes in his working manual [29]: geocentric, conformal, authalic, parametric, rectifying, and isometric. The first five of these auxiliary latitudes were systematically described by O. Adams, who derived all the corresponding formulas in 1921 [30], but they gained popularity much later, after the publication of Snyder's manual.

The basic latitude used in global datasets and position determination based on global navigation is *geodetic latitude* ($\theta$). It represents the angle between the equatorial plane and the surface normal at a point on the ellipsoid. Calculating geocentric latitude ($\phi$), which represents the angle between the equatorial plane and the radius vector, is relatively simple compared to other auxiliary latitudes. Equation (1) can be used to calculate the geocentric latitude based on the geodetic latitude, where $e$ denotes the eccentricity of the ellipsoid.

$$\varphi = \arctan\left((1 - e^2) \cdot \tan(\theta)\right) \tag{1}$$

Two auxiliary latitudes are of significant importance in addressing specific types of distortion. The application of the conformal latitude ($\chi$) results in conformal mapping of an ellipsoid onto a sphere, effectively eliminating angular distortion. On the other hand, the use of the authalic latitude ($\beta$) achieves equal-area mapping, eliminating areal distortion. The conformal latitude can be computed from the geodetic latitude using Equation (2).

$$\chi = 2 \cdot \arctan\left(\tan\left(\frac{\pi}{4} + \frac{\theta}{2}\right) \cdot \left(\frac{1 - e \cdot \sin(\theta)}{1 + e \cdot \sin(\theta)}\right)^{\frac{e}{2}}\right) - \frac{\pi}{2} \tag{2}$$

Computing the geodetic from the conformal latitude, i.e., the inverse transformation, requires an iterative procedure or series [29,30]. Equation (3) is one of the methods for the inverse transformation. By using the first four terms of the sum, the computational error can be kept below $10^{-12}$. The corresponding values for the coefficients $c_i$ can be found in [29].

$$\theta = \chi + \sum_{i=1}^{\infty} c_i \cdot \sin(2\chi) \tag{3}$$

The authalic latitude is calculated from the geodetic latitude using Equations (4)–(6).

$$q = \left(1 - e^2\right) \cdot \left(\frac{\sin(\theta)}{1 - e^2 \cdot \sin^2(\theta)} - \frac{1}{2 \cdot e} \ln\left(\frac{1 - e \cdot \sin(\theta)}{1 + e \cdot \sin(\theta)}\right)\right) \tag{4}$$

$$q_p = q_{\theta=90°} = \left(1 - e^2\right) \cdot \left(\frac{1}{1 - e^2} - \frac{1}{2 \cdot e} \ln\left(\frac{1 - e}{1 + e}\right)\right) \tag{5}$$

$$\beta = \arcsin\left(\frac{q}{q_p}\right) \tag{6}$$

Similar to the conformal latitude, the inverse transformation for the authalic latitude requires an iterative procedure or series. Figure 3a depicts the deviation of the aforementioned auxiliary latitudes from the geodetic latitude. It can be observed that the geocentric and conformal latitudes have very similar deviations from the geodetic latitude. Figure 3b displays the difference between the conformal and geocentric latitude values as a function of geodetic latitude. The maximum deviation occurs at 60° north and south (geodetic) latitude and is approximately $1.4 \times 10^{-2°}$ (50.4″). Considering the small deviation from conformal latitude, the relative simplicity of the calculation, and the availability of a straightforward

closed-form inverse transformation, geocentric latitude can be used to mitigate the angular distortion of mapping an ellipsoid onto a sphere.

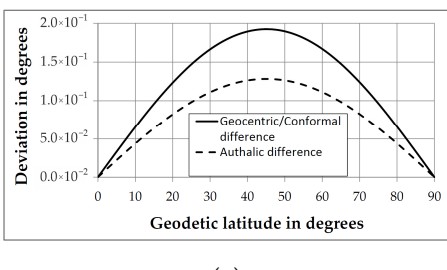

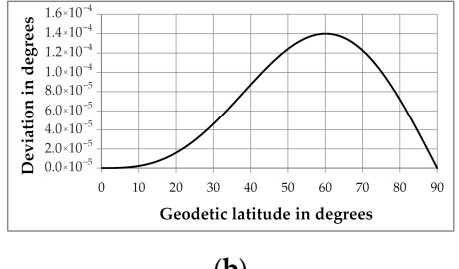

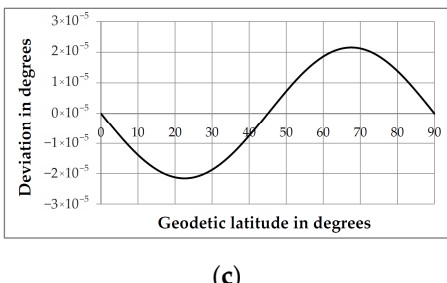

(a)

(b)

(c)

**Figure 3.** Deviations: (**a**) auxiliary latitudes from geodetic latitude ($\theta - \phi$, $\theta - \chi$, and $\theta - \beta$); (**b**) conformal from geocentric ($\chi - \phi$); (**c**) approximated authalic from authalic ($\beta - \beta'$).

To simplify the calculation of authalic latitude and gain closed-form inverse transformation, we propose the following approximated formula:

$$\beta\prime = \arctan\left(\left(1 - e^2\right)^k \cdot \tan(\theta)\right) \qquad (7)$$

The smallest maximum deviation of the approximation ($\beta'$) from the authalic latitude ($\beta$) for the WGS 84 ellipsoid, obtained by an iterative process of checking the maximum values of the function for various values of k, is achieved with $k = 0.666741$ and is smaller than $2.16 \times 10^{-5\circ}$ (0.078″), as shown in Figure 3c. This represents a 33% improvement over the approximation presented in [31]. The proposed formula is similar to the geocentric latitude formula, and both the forward and inverse transformations have closed forms and can be easily computed. The ratio of the tangents of the geodetic latitude to the approximated authalic latitude for the WGS 84 ellipsoid is approximately 1.004488.

### 3.2. Mapping a Sphere onto a Set of Planes

While a sphere is more suitable for geodesic calculations compared to an ellipsoid, it cannot be flattened into a plane without interruptions. To overcome this limitation, the next step involves projecting the sphere onto another figure that has flat surfaces or can be unrolled into a plane seamlessly. GDGGSs achieve this by utilizing the faces of circumscribed regular or semi-regular polyhedra as the projection surfaces for the sphere.

Each face of the polyhedron represents a partition of a specific projection. Platonic solids are commonly used as the base polyhedra [4] because they possess regularity, with faces of the same shape (triangles, squares, or pentagons), equal areas, and an equal number of neighboring faces. Among the regular polyhedra, the hexahedron (cube) is particularly popular due to its relatively small number of partitions (six) and the square shape of both the partitions and cells.

In addition to regular polyhedra, a commonly used semi-regular polyhedron is the truncated icosahedron [32], which belongs to the group of 14 Archimedean solids. The truncated icosahedron lends itself to a hexagonal cell structure, although its sides are not all identical. It consists of 12 pentagonal and 20 hexagonal sides.

Cylindrical projections have low distortion along a freely chosen pseudoequator [33], so they can represent very long areas with moderate distortion. One of the oldest and simplest cylindrical projections still in use today is equidistant cylindrical projection. It was invented by Marinus of Tire around 100 AD and, despite its considerable distortion, remains the most commonly used projection for organizing global data. This projection establishes a straightforward relationship between map positions and the corresponding geographic locations.

By combining two orthogonal equidistant cylindrical projections, a projection known as Yin–Yang [6] is obtained. The Yin–Yang projection, together with its associated grid, has

found applications in a variety of fields, such as simulations of geodynamo and mantle convection [6], visualization of 3D mantle convection [34], global shallow water models [35], 3D hydrodynamic simulations of core-collapse supernova evolution [36], feature extraction from omnidirectional panoramic images [37], and many others. However, its potential as a cartographic projection has not been extensively explored.

In general, the Yin–Yang approach uses two complementary components, and the mapping need not be based on orthogonal equidistant cylindrical projections. However, for simplicity, it is usually implemented in this way. To refer to the projection more precisely in this paper, we use the term Dual Orthogonal Equidistant Cylindrical (DOEC) projection. The name is proposed according to the modern classification of map projection [38]. According to the type of distortion, DOEC is an equidistant projection since the local linear scale factor along one of the main directions is equal to one. According to the shape of the pseudograticule, it is a cylindrical projection that represents pseudomeridians as mutually parallel straight lines and pseudoparallels as mutually parallel straight lines perpendicular to the pseudomeridians.

The first partition (P0), in the normal aspect, extends along the equator and is symmetrical about the equator and the prime meridian. It uses polar coordinates identical to global geographic coordinates ($\varphi$, $\theta$). The second partition (P1), in the transverse aspect, extends along the anti-meridian (180th meridian), is symmetric about it, and includes both poles. Figure 2d shows the two partitions in a rectangular shape that facilitates the determination of their boundaries, as defined in the logical expression (10). In this shape, however, the partitions overlap by about 6.4% of their total area. Figure 4 shows the partitions without overlaps and their distribution over the globe. The non-overlapping shapes of the partitions shown in Figure 4 result from the elimination of cells that satisfy condition (11). Due to the rectangular shape of the sections and cells, there is still some overlap at the edges of the partitions, but it decreases with increasing resolution and tends to zero.

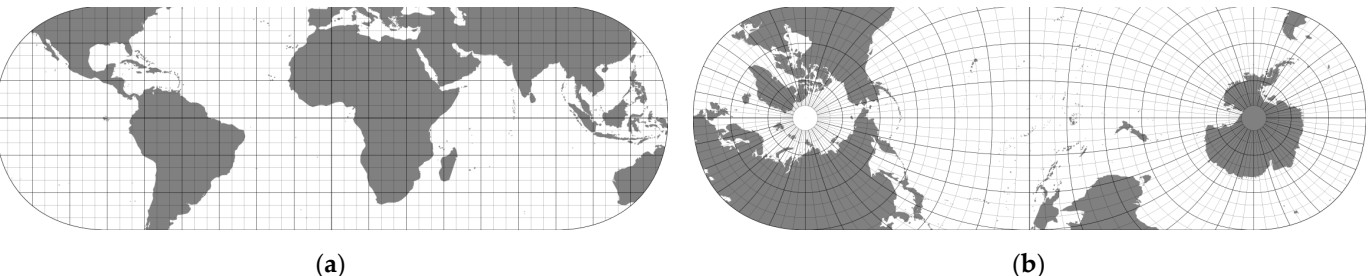

(**a**)  (**b**)

**Figure 4.** Two complementary partitions P0 (**a**) and P1 (**b**) of DOEC projection, respectively, in the normal and transverse aspects, and with no overlapping areas. The extent of the land mass and the graticule are displayed.

The local coordinate system of partition P0 coincides with the global geographic grid, so no coordinate conversion is required. On the other hand, the local polar coordinates of partition P1 are determined based on either the geographic coordinates or the coordinates of partition P0 using Equations (8) and (9).

$$\theta_{P1} = \arcsin(-\cos(\theta_{P0}) \cdot \sin(\phi_{P0})) \tag{8}$$

$$\phi_{P1} = -\text{sgn}(\theta_{P0}) \cdot \arccos\left(\frac{-\cos(\theta_{P0}) \cdot \cos(\phi_{P0})}{\cos(\theta_{P1})}\right) \tag{9}$$

Because of the orthogonality of the partitions, Formulas (8) and (9) can also be used to convert coordinates from partition P0 to P1 by simply exchanging the arguments. The condition indicating that a point with local polar coordinates ($\varphi_p$, $\theta_p$) belongs to the current rectangular partition, including overlapping areas, is defined by the logical expression (10).

The expression (11) additionally indicates that the point belongs to the overlapping area, where $\theta_q$ is the latitude in the complementary partition obtained by Equation (8).

$$\left(-\frac{3\pi}{4} < \phi_p < \frac{3\pi}{4}\right) \wedge \left(-\frac{\pi}{4} < \theta_p < \frac{\pi}{4}\right) \tag{10}$$

$$\left(\phi_p < -\frac{\pi}{2} \wedge \theta_q < \frac{\pi}{4}\right) \vee \left(\phi_p > \frac{\pi}{2} \wedge \theta_q > -\frac{\pi}{4}\right) \tag{11}$$

### 3.3. Rotation of the Graticule System

The distribution of the distortion depends on the projection applied and is never uniformly distributed over the surface of the partition. Typically, the distortion is minimal in the center of the partition and increases toward the edges and corners, indicating a greater distance of the projection plane from the surface of the sphere.

Tissot's indicatrices [29] are commonly used to visualize distortions. They are represented as ellipses formed by projecting infinitesimal circles from the surface of the globe onto the projection plane. The size, eccentricity, and inclination of these ellipses indicate the type and degree of distortion present.

To better observe the distribution of the deformation parameters, instead of using ellipses that combine multiple deformation parameters graphically, we represent each parameter individually using a red color intensity scale. The values of the distortion parameters are still calculated using the indicatrices. For instance, the angular distortion is determined by calculating the maximum angular deformation $\omega$ using Equation (12) based on the major (*a*) and minor (*b*) semi-axes of the indicatrix. The formulas for calculating the indicatrices and the corresponding deformation parameters can be found in [29].

$$\omega = 2 \cdot \arcsin\left(\frac{|a-b|}{a+b}\right) \tag{12}$$

Figure 5 illustrates the distribution of angular, areal, and aspect distortion for both DOEC partitions for the basic orientation of the graticule system, as described in the previous section. The aspect distortion refers to the ratio of the major and minor semi-axes of the indicatrix (*a*/*b*). Due to the property of DOEC projection and the fact that *b* is always equal to 1, the values for the surface and aspect distortion are identical.

Changing the projection aspect by rotating the graticule system does not affect the distribution of the distortion [33], which can be used to reduce the deformation of the region of interest. In cartography, adjusting the position of projection planes or rotating circumscribed polyhedra [39] are common techniques to achieve certain desired effects. Even in the oldest hexahedral projection [40], the base cube is rotated 27° about Earth's axis of rotation to align the westernmost point of the African continent with the edge of the cube. An early discussion of the orientation of the base cube can be found in [8], but without an in-depth examination of the underlying considerations and applications.

The main reasons for the change in the basic orientation, which assumes alignment with the equator, the prime meridian, and the poles, can be summarized as follows:

- Avoiding fragmentation of target areas: Adjusting the orientation helps prevent splitting local or regional target areas across multiple faces of the polyhedron [41]. This ensures the integrity of these areas in the projection.
- Encompassing an entire continent: Changing the orientation allows an entire continent, such as North America, to be included in a single partition [8,32]. This is beneficial for regionally focused mapping and analysis and is suitable for a polyhedron with fewer and larger faces.
- Preventing ruptures in the continental plates after the base polyhedron has unfolded: This is achieved by positioning the vertices of the polyhedron at the oceans, as demonstrated in Fuller's Dymaxion Airocean World Map [42].

- Minimizing landmass distortion: Another important criterion for the orientation is minimizing landmass distortion [31]. This aims to preserve the accurate representation of land features on the map.

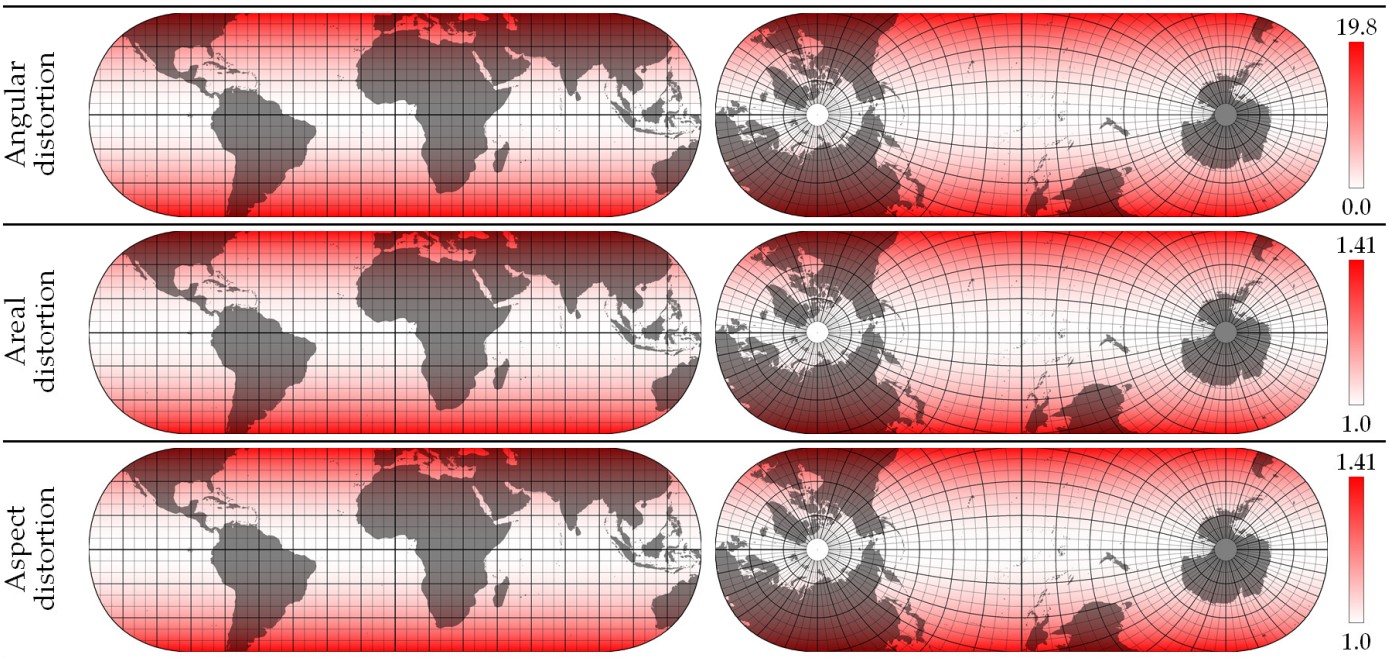

**Figure 5.** The distribution of angular, areal, and aspect distortion for both DOEC partitions for the basic orientation of the graticule system.

The first two reasons mentioned above are location-specific and may not be applicable to global systems since they prioritize specific regions. On the other hand, the latter two reasons are more universal and serve as criteria for determining the orientation used in DOEC. However, both criteria cannot be met at the same time. So, we first used an iterative process and a rotation around all three Cartesian axes with a check of angular and areal distortions to determine the optimal orientation considering landmass distortion.

To determine the optimal orientation, a vector map of the world [43] was rasterized in LatLon WGS 84 (EPSG:4326) projection at a resolution of $4096 \times 2048$ pixels (see Figure 6). The map was then reprojected on-the-fly into two partitions with progressively varying rotation angles. Nearest neighbor sampling was used in the reprojection since it is fast and clearly delineates the continents. In addition, Antarctica was excluded from the map to focus on more cartographically important areas. The distortion is checked only for the cells that belong to the landmass.

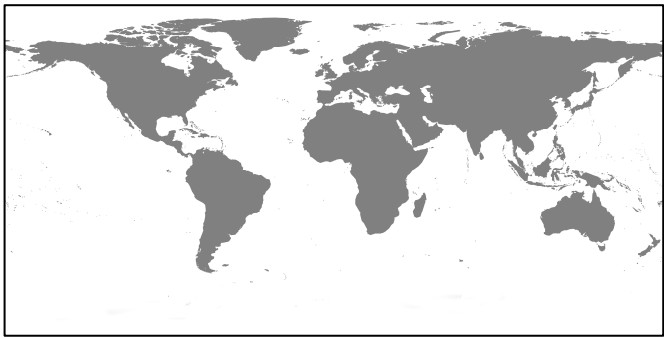

**Figure 6.** Rasterized world map without Antarctica in LatLon WGS 84 (EPSG:4326) projection used to test land mass distortion.

Considering the rotation angles, denoted as $\varphi_r$ for the vertical axis (longitudinal rotation), $\theta_r$ for the horizontal axis (latitudinal rotation), and $\rho_r$ for counterclockwise rotation about the axis perpendicular to the previous two, the minimum distortion of the landmass at one degree resolution was obtained as $\varphi_r = 131°$, $\theta_r = 49°$, and $\rho_r = -20°$. To further reduce the clipping of the continental plates, the following corrected rotation angles are proposed: $\varphi_r = 125°$, $\theta_r = 50°$, and $\rho_r = -15°$. Figure 7 illustrates the layout of the partitions based on the proposed rotation angles. The advantages of DOEC projection and the effects of the proposed method on distortion reduction are shown in the next section.

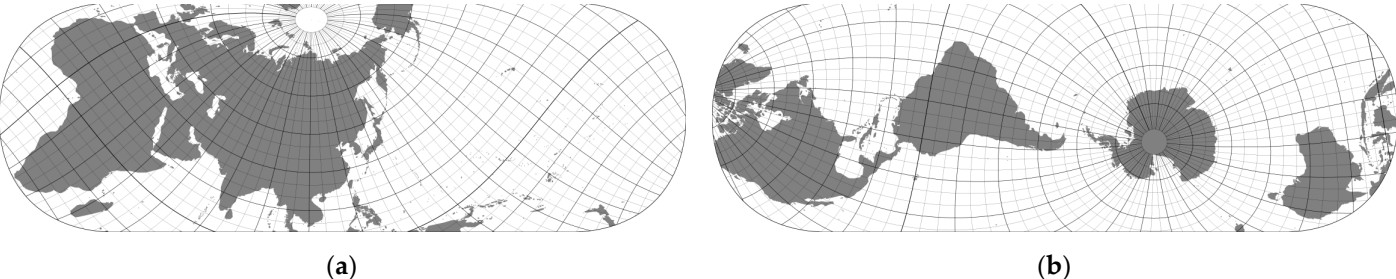

(**a**)  (**b**)

**Figure 7.** Two complementary partitions P0 (**a**) and P1 (**b**) of DOEC projection obtained for optimally rotated graticule system ($\varphi_r = 125°$, $\theta_r = 50°$, and $\rho_r = -15°$) to minimize landmass distortions and continental ruptures.

Since the effect of distortion increases with distance from the pseudoequator, the most unfavorable regions are located near the edges of the partitions. In the case of the proposed graticule rotation, the distortion is greatest in the region of northwestern Europe (Ireland and Great Britain), Alaska, the Indonesian archipelago, and New Zealand.

Optimizing the projections by rotating the graticule system leads to more complex data preparation that requires additional transformations. In addition, maps using this rotated graticule system may present a challenge to users accustomed to traditional north-oriented maps that adhere to long-standing cartographic conventions. While it is possible to selectively use map sections by applying an additional rotation that aligns the area of interest to the north, it is important to emphasize that the main goal of the proposed projection is to efficiently organize geospatial data to minimize distortion and facilitate three-dimensional visualization.

## 4. Experimental Results and Discussion

DOEC projection offers several key advantages over circumscribed polyhedra, the typical choice in DGGSs. First, it reduces the number of partitions to only two. This property proves beneficial for out-of-core terrain rendering algorithms, such as Ellipsoidal Clipmaps [5]. Minimizing the number of partitions displayed simultaneously reduces memory consumption since each partition requires corresponding structures for visualization. These structures typically include terrain elevations and high-resolution aerial imagery at multiple levels of detail. For out-of-core algorithms, the constant updating of these structures with geospatial data places a burden on the central processor and requires access to network resources or slow secondary media to retrieve the data, so the number of partitions directly affects system performance.

In addition, partition connections require special treatment in three-dimensional visualization, which includes additional testing of conditions, clipping, and fitting. Therefore, minimizing the occurrence of partition connections improves overall performance. Moving the partition boundaries above the water surface further facilitates seamless joining. Consequently, the proposed DOEC projection incorporates a rotated graticule system to not only reduce distortion but also minimize clipping of the continental plates by the partition boundaries. In visualizations, the water surface is usually represented dynamically by generating details using procedural techniques. This means that the details are created



algorithmically instead of relying on real geospatial data. Therefore, discontinuities or irregularities at the edges of the partitions overlying the waters do not affect the generation of the visual scene. However, if it is necessary to analyze ocean basins specifically, it would be better to use a different oblique aspect of the projection and superimpose the boundaries of the partitions over the continents.

Another advantage of DOEC projection is its ability to achieve a favorable balance between areal and angular distortions. Table 1 shows the distortion values for common projections considering a perfect sphere and without rotations of the projection planes. These values were derived from more than 67 million measurement points evenly distributed over the surface of the partitions.

**Table 1.** The comparison of angular ($\omega$), areal ($\sigma$), and aspect ($\alpha$) distortions for the following projections: Quadrilateralized Spherical Cube (QSC), revised Hierarchical Equal-Area isoLatitude Pixelization (rHEALPix), Adjusted Spherical Cube (ASC), Continuous Cube Mapping (CCM), Cartesian Spherical Cube (CSC), and Dual Orthogonal Equidistant Cylindrical (DOEC). rHEALPix projection parameters are presented independently for equatorial (rHEALPix$^E$) and polar (rHEALPix$^P$) regions. In addition to the minimum (min), maximum (max), and average (ave) values of the distortion parameters, the geometric mean of the $\alpha_{ave}$ and $\sigma_{ave}$ values (GM$_{\alpha\sigma}$) is also displayed.

| Projection | Angular Distortion | | | Areal Distortion | | | Aspect Distortion | | | $GM_{\alpha\sigma}$ |
|---|---|---|---|---|---|---|---|---|---|---|
| | $\omega_{min}[°]$ | $\omega_{max}[°]$ | $\omega_{ave}[°]$ | $\sigma_{min}$ | $\sigma_{max}$ | $\sigma_{ave}$ | $\alpha_{min}$ | $\alpha_{max}$ | $\alpha_{ave}$ | |
| QSC | 0.0 | 25.081 | 16.129 | 1.0 | 1.0 | 1.0 | 1.0 | 1.555 | 1.331 | 1.154 |
| rHEALPix$^E$ | 0.0 | 24.107 | 7.964 | 1.0 | 1.0 | 1.0 | 1.0 | 1.528 | 1.155 | 1.075 |
| rHEALPix$^P$ | 13.807 | 49.250 | 31.320 | 1.0 | 1.0 | 1.0 | 1.273 | 2.429 | 1.770 | 1.330 |
| ASC | 0.0 | 31.084 | 11.572 | 1.0 | 1.414 | 1.187 | 1.0 | 1.732 | 1.234 | 1.210 |
| CCM | 0.0 | 31.084 | 9.078 | 1.0 | 2.083 | 1.344 | 1.0 | 1.732 | 1.179 | 1.259 |
| CSC | 0.0 | 31.087 | 11.489 | 1.0 | 1.333 | 1.104 | 1.0 | 1.732 | 1.235 | 1.168 |
| DOEC | 0.0 | 19.759 | 5.864 | 1.0 | 1.414 | 1.113 | 1.0 | 1.414 | 1.113 | 1.113 |

DOEC is compared to other projections that also use square cells and can be treated as hexahedral projections. These include Quadrilateralized Spherical Cube (QSC), rotated Hierarchical Equal-Area isoLatitude Pixelization (rHEALPix), Adjusted Spherical Cube (ASC), Continuous Cube Mapping (CCM), and Cartesian Spherical Cube (CSC).

Further, rHEALPix is a hybrid projection that combines Lambert cylindrical equal-area projection for the equatorial region with interrupted Collignon projection for the polar regions. To distinguish between these two regions, they are referred to as rHEALPix$^E$ and rHEALPix$^P$, respectively, in Table 1. Both the QSC and rHEALPix$^P$ projections are equal-area projections but have significant angular distortions and even discontinuities along the diagonals of the partitions. This is due to the fact that their partitions consist of four triangular surfaces. The other three projections, on the other hand, offer a more balanced compromise between surface distortion, angular distortion, and ease of implementation.

Figure 8 provides a visual comparison of the projections listed in Table 1, offering insights into the distortion effects. Although rHEALPix$^P$ projection is primarily designed for the polar regions, it is applied to a portion of the equatorial region (Figure 8f) to demonstrate the distortion effects on similar shapes compared to the other projections. To quantify the combined effect of aspect and areal distortions, Table 1 contains the geometric mean of the average values of aspect and areal distortions calculated using Equation (13).

$$GM_{\alpha\sigma} = \sqrt{\alpha_{ave} \cdot \sigma_{ave}} \tag{13}$$

Table 1 shows that DOEC projection has the lowest values for maximum and average angular distortion, maximum and average aspect distortion, and relatively low average area distortion. Consequently, it achieves the second lowest $GM_{\alpha\sigma}$ value, just after rHEALPix$^E$.

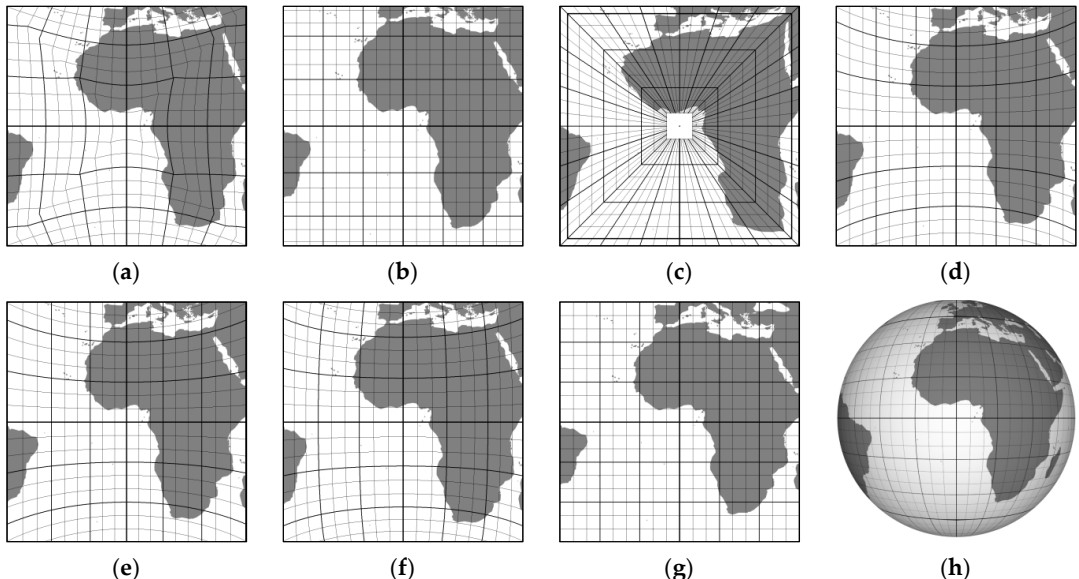

**Figure 8.** Graphical comparison of projections whose distortion parameters are listed in Table 1: (**a**) QSC; (**b**) rHEALPix$^E$; (**c**) rHEALPix$^P$; (**d**) ASC; (**e**) CCM; (**f**) CSC; (**g**) DOEC; and (**h**). The three-dimensional shape of the continents and the graticule.

Another measure of projection quality based on Tissot's indicatrix is the grid over-sampling factor (*GOF*) [18]. It estimates the local oversampling of the data that occurs when projecting general satellite imagery onto a regular grid. The *GOF* is defined by Equation (14), where $a$ and $b$ are the semi-axes of the indicatrix and $b_{min}$ is the global minimum value for the minor semi-axis across the entire partition.

$$GOF = \frac{a \cdot b}{b_{min}^2} \qquad (14)$$

In DOEC projection, $b_{min}$ is equal to 1, so *GOF* reduces to $a \cdot b$. Moreover, when applied to a perfect sphere, DOEC has a constant value for $b$ over the entire partition and is equal to 1, so the value for *GOF* reduces only to the value of $a$. Since aspect distortion is calculated as $a/b$ and $b$ is equal to 1, the values for DOEC and aspect distortion are identical. Taking this into account and based on Table 1, the *GOF* for DOEC is 1.113, which means that 11.3% more data samples than the theoretical minimum must be stored. For an equidistant cylindrical projection with a single partition (plate carree), the *GOF* is 1.81, while it decreases to 1.36 when the latitude is restricted to the range $[-56°, 72°]$, as shown in [18].

Table 2 provides insight into the effects of graticule system rotation on average distortions for some characteristic values. The values listed in the table were obtained by averaging over 6.7 million measurement points belonging to the landmass. The location of the landmass was determined by reprojecting a rasterized world vector map [43] using the appropriate rotation of the graticule system. Applying the rotations $\varphi_r = 125°$, $\theta_r = 50°$, and $\rho_r = -15°$ (hereafter abbreviated as R(125°, 50°, −15°)) reduces the average angular distortion of the landmass by 1.9 times and the average areal distortion by about 6%. The experiments were performed with different rotational steps, from 90° to 1°, with smaller steps in ranges of values where larger steps provided good results and where minima could be expected.

The rotation R(131°, 49°, −20°) yields the lowest average value of angular distortion but causes the partition boundary to intersect the southern part of the African continent (Figure 9), resulting in a reduction in average angular distortion of less than 1%. Considering the optimization of the disruptions in the continental plates, the rotation R(125°, 50°, −15°) proves to be a superior solution. It effectively reduces the ruptures in the continental plates while providing a notable improvement in the average distortions.

**Table 2.** The comparison of the averaged angular ($\omega_{ave}$), areal ($\sigma_{ave}$), and aspect ($\alpha_{ave}$) distortions for the normal aspect of the projection (R(0°, 0°, 0°)), the proposed optimal oblique aspect (R(125°, 50°, −15°)), and the oblique aspect with the minimum landmass distortion (R(131°, 49°, −20°)).

| $R(\varphi_r, \theta_r, \rho_r)$ | $\omega_{ave}[°]$ | $\sigma_{ave}$ | $\alpha_{ave}$ |
|---|---|---|---|
| R(0°, 0°, 0°) | 6.721 | 1.130 | 1.130 |
| R(125°, 50°, −15°) | 3.557 | 1.067 | 1.067 |
| R(131°, 49°, −20°) | 3.523 | 1.067 | 1.066 |

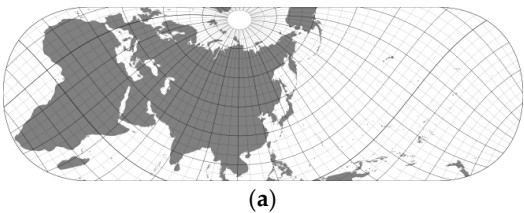 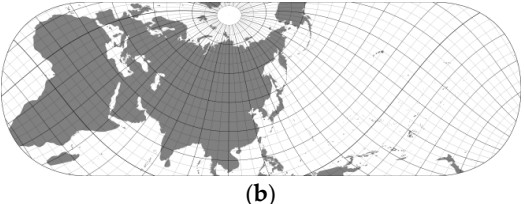

(**a**)  (**b**)

**Figure 9.** Comparison of the appearance of the partitions P0 for the oblique aspects with graticule system rotations: (**a**) R(125°, 50°, −15°); (**b**) R(131°, 49°, −20°).

The optimal oblique aspect of the projection reduces the GOF to 1.067, which is still higher than the GOF for the Equi7 grid based on equidistant cylindrical projections, where it is 1.03 for Africa and Asia [18]. However, considering that the DOEC partitions cover a much larger area than the Equi7 partitions and are not centered with respect to the continental plates for which the distortion is calculated, the obtained result can be considered comparable.

Table 3 presents a comparison of the distortion parameters for different auxiliary latitudes applied to the entire surface of partition P0 without rotating the graticule system. The results show that applying conformal latitude to map an ellipsoid onto a sphere yields identical values for angular and aspect distortion as for an ideal sphere (see first and second rows in Table 3), indicating that this mapping does not result in any additional angular distortion. The application of the authalic latitude also shows no additional areal distortion. The average angular distortion deviation when using the geocentric latitude is about 0.0026%, while the average areal distortion using approximate authalic latitude is less than 0.00018%. These results indicate that the use of approximated auxiliary latitudes does not introduce significant additional distortion. It is worth noting that the use of approximated authalic latitude even reduces the average angular distortion in this particular case. Its simplicity and favorable performance make it the optimal choice for mapping the ellipsoid onto the sphere.

**Table 3.** Comparison of the effects of applying different auxiliary latitudes on the distortion of the P0 partition.

| Latitude | Angular Distortion | | Areal Distortion | | Aspect Distortion | |
|---|---|---|---|---|---|---|
| | $\omega_{max}[°]$ | $\omega_{ave}[°]$ | $\sigma_{max}$ | $\sigma_{ave}$ | $\alpha_{max}$ | $\alpha_{ave}$ |
| Sphere | **19.758564** | **5.864603** | **1.414214** | **1.113448** | **1.414214** | **1.113448** |
| Conformal | **19.758564** | **5.864603** | 1.418974 | 1.114840 | **1.414214** | **1.113448** |
| Geocentric | 19.758882 | 5.864757 | 1.418971 | 1.114838 | 1.414222 | 1.113451 |
| Authalic | 19.695547 | 5.774060 | **1.414214** | **1.113448** | 1.412636 | 1.111693 |
| Approx. authalic | 19.695632 | 5.774065 | **1.414**209 | **1.113**446 | 1.412638 | 1.111693 |

All the previously presented results indicate that the proposed mapping of Earth's geospatial data can be effectively used to mitigate distortions and organize the data into only two partitions.

## 5. Conclusions

Dual Orthogonal Equidistant Cylindrical (DOEC) projection offers several advantages over other projections in terms of reducing memory consumption and improving overall performance for world-scale geospatial data visualization, minimizing partition interconnections, and achieving a favorable balance between areal and angular distortions. Compared to other projections, such as Quadrilateralized Spherical Cube (QSC), rotated Hierarchical Equal-Area isoLatitude Pixelization (rHEALPix), Adjusted Spherical Cube (ASC), Continuous Cube Mapping (CCM), and Cartesian Spherical Cube (CSC), DOEC projection has lower values for maximum and average angular distortion, maximum and average aspect distortion, and relatively low average area distortion. It also incorporates a rotated graticule system to minimize landmass distortion and continental plate disruption. The optimal oblique aspect of the projection reduces the average angular landmass distortion to about 3.6°, the average area distortion to about 1.07, and the grid oversampling to about 6.7% while optimizing continental plate disruptions. Furthermore, by applying the approximated authalic latitude, DOEC projection preserves the areal distortion while additionally reducing the average angular distortion and is considered the best candidate for mapping an ellipsoid onto a sphere. Overall, DOEC projection provides improved performance and distortion characteristics, making it a valuable choice for mapping global geospatial data.

**Author Contributions:** Conceptualization, Aleksandar Dimitrijević; methodology, Aleksandar Dimitrijević; validation, Aleksandar Milosavljević; writing—original draft preparation, Aleksandar Dimitrijević; writing—review and editing, Aleksandar Dimitrijević and Aleksandar Milosavljević; visualization, Aleksandar Dimitrijević; supervision, Dejan Rančić; funding acquisition, Dejan Rančić. All authors have read and agreed to the published version of the manuscript.

**Funding:** This research was supported by Ministry of Education, Science and Technological Development of the Republic of Serbia, grant number 451-03-47/2023-01/200102.

**Data Availability Statement:** No other data than those available in the text were created or analyzed in this study. Data sharing is not applicable to this article.

**Conflicts of Interest:** The authors declare no conflict of interest.

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
