# Peer review of "Efficient Distortion Mitigation and Partition Reduction in Mapping Global Geodata: Dual Orthogonal Equidistant Cylindrical Projection Approach"

_ijgi, doi:10.3390/ijgi12070289_

Round 1

Reviewer 1 Report

The authors highlight in their study an extremely relevant topic for the global geospatial community which is, despite its fundamental importance to geospatial data, unfortunately still widely underrepresented in the scientific literature of the domain. They provide novel ideas for optimising solutions and benchmarking for global geospatial grids in a manuscript which is overall very well written and illustrated. It adds considerable insight on possible future global grid systems and is definitely worth publishing.

While the authors start the paper arguing for DGGS as a concept to organise 'big' geospatial data, when proposing DOEC they practically leave that path. DOEC will have to have overlaps in order to avoid gaps. What is missing in the discussion is how the yin-yang can be utilized for efficient hierarchical multi-scale spatial partitioning, or in other words why it should be benchmarked against DGGS which have this one unique advantage (no gaps & no overlaps). (See also comments on lines 49-52 below)

While exploring whether and how DOEC and DGGS can be further integrated might lead beyond the scope of this paper, the analysis at this stage would benefit from a benchmarking against other single or multi-partition global projection systems frequently used for global geospatial data like (pseudo-) Plate-Caree (1 partition but singularities and huge distortions) or UTM (low distortions but 60 partitions) as the two extremes and Equi7 as an example of a compromise [1]. Especially the latter should be discussed in this context and the reference might be a valuable addition to the paper as it well explains the role of distortions and metrics how to benchmark it.

[1] Bauer-Marschallinger et al. (2014) Optimisation of global grids for high resolution remote sensing data, Computers & Geosciences, 72, 84-93

Further comments referenced to specific lines of the manuscript:

49-52 please be a bit more specific, it is clear that a change in partition is detrimental to ‘seamless’ analysis, but the effects are not the same depending on whether just the projection changes or also the grid boundaries with respective gaps or overlaps. Maybe distinguish between effects on rendering (visualisation) and those on data analysis for which it is important that the grid is contiguous (and can thus remain unchanged) avoiding resampling when going across partitions.

132: the term ‘acquisition’ might be a bit misleading here (in EO means the acquisition of observations by sensors’). Is what is meant here the ‘indexing and addressing’ of DGGS, or does it refer to ‘retrieval’ meaning ‘identifying and extracting’ specific parts of the data, which goes a bit beyond?

148: is there a reference or a quick explanation why this is the case?

150: and no overlaps!

233: polyhedral -> polyhedra

304/305: maybe revise sentence, unclear (to me) what is meant by ‘surface’ distortion, is it ‘area’ distortion? Why is it related to aspect distortion?

340: these grid cells (better than ‘pixel’) are not equal area as higher latitudes receive many more (smaller) cells, does that affect analysis?

367: Maybe explain a bit more detailed what a ‘partition’ is. Why are faces of a hexahedron, if concatenated not a single partition?

374: Shouldn’t partition connections only require this kind of attention if the boundaries of the partition do not coincide with the boundaries of the cells (which is the great advantage of the hexahedron)?

378: Why is seamless joining over water easier?

394: Why 67 million points?

483: isn’t DOEC applying approximated authalic latitudes in the end mapping an ellipsoid onto a cylinder?

Reviewer 2 Report

It is not clear how to calculate k = 0.666741 for another ellipsoid

It is good to make comparison of presentation of the World not only with projections on Figure 8 but also with classical projections like Cylindrical Equal-Area Projection Oblique Case and others

Reviewer 3 Report

This is a very good paper! The use of English language is better than in average papers. I reccommend it for acceptance.

The paper reduces distortion of a previously developed map projection by using the oblique aspect. Aspect parameters are determined by numerical optimization.

Not reducing the merits of the paper, it has two serious flaws. Both should be corrected quite easily:

Problem #1: The study does not use units of measurements at all. For example, in table 1, omega seems to be listed in degrees, but there is no degree sign. The reader might interpret the values in radians. Please always indicate the units you use, especially because degrees are not in SI.

Formula 13 also shows the lack of proper use of units. The authors multiply degrees with a dimensionless ratio, and take its root. In what unit they recieved results? Square-root of degrees? Does not seem to be a useful quantity. Instead of omega, I would recommend using alpha to consider angular distortion. This would result in a meaningful dimensionless number.

Before correcting this issue, please read DOI 10.1080/15230406.2020.1768439 to get a deep insight into the problem.

Problem #2: The paper uses obsolete, out-of-date terminology of map projections found in old literature. The usage of old terms, fortunately, did not influence the validity of results, but make the paper look "less scientific" and "childish". This could be corrected quite easily by a search&replace.

Before proceeding, please read these three papers:
10.3390/geographies2020019
10.1080/23729333.2016.1184554
https://archive.org/details/sevenaspectsofge0011wray/

Now you can see that in recent paradigm of map projections, cylindrical projections are not defined as projections onto a cylindrical surface (as this definition would lead us to false conclusions), but as map projections onto a plane, in which parallels and meridians are perpendicular straight lines.

Instead of rotating a cylinder, we rather rotate the graticule system on the globe to achieve the oblique aspect. Please use the terms related to metagraticule or pseudograticule (both nomenclatures are accepted by the community).

The following terms: "cylindrical surface", "projection cylinder", "Orientation of projection cylinders" are all deprecated, and should be avoided.

The phrases "If the projection surfaces are not flat, they are unfolded", "or can be unrolled into a plane seamlessly" and "Cylinders are often used as an alternative to flat surfaces for projections." contain references to out-of-date terminology, and should be deleted. Figure 2/c should be deleted or replaced.

The claim "Due to its better adhesion to the spherical surface, the cylinder produces less distortion even with a smaller number of projection surfaces." is false, and is one of the reasons why you should really avoid old terminology. For example, the best map projection of a hemisphere is azimuthal (DOI 10.2307/2317182). Please write instead: "Cylindrical projections have low distortion along a freely chosen metaequator/pseudoequator [pick which you like more], so they can display very long areas with moderate distortion."

Minor comments:

Please always write equal-area with hyphens. (Now most occurences are hyphenated correctly, but not all.)

You defined auxiliary latitudes using a projection onto an auxiliary sphere. This is the proper definition for rectifying, conformal, and authalic latitudes. In spite, geocentric, parametric, and isometric latitudes are defined on the ellipsoid without projecting onto any sphere. Strictly speaking, the isometric one is not even a proper latitude, as it is not an angle in [-90°;90°].

Otherwise, congratulations!

Reviewer 4 Report

The authors present a novel global map projection system based on the yin‐yang design. The study is overall intriguing and in proper English. In the following, I provide a brief assessment.

1) The study aims for providing a suitable and performant reference system for massive geospatial data from high-resolution satellite data. In this context, I find it very irritating that the study discusses the work only in relation to common and specialised polyhedral designs, and completely leaves out already existing and used projected grid systems as the UTM, Equi7, or EASE.

2) Results in Figure 5 surprised me, as the regular Equirectangular projection (or the Plate Carree) has zero distortions at the equator, with Tissot indicatrices as perfect circles at unity size. So: Why does the areal distortion map show minimum values of 1.62 (at the equator), and not 1.0? Perhaps this stems from introducing auxiliary latitudes, but the difference to 1.0 is very large...

3) The yin‐yang approach (although the work of another study) is very intriguing, with just two cylindrical projections to map the entire globe. I'm wondering, in the search for optimal orientation to cover and not-split the land surface, are there other local mimima than the presented one in Figure 7? (which is nice, though)

overall fine.

Round 2

Reviewer 4 Report

Thank you for the rapid response and that many of the reviewers' comments are already integrated. I still have major issues to raise:

1) major

I see serious troubles with the values for areal scale factor plotted in Fig5 and plotted in Tab1. The areal distortion for the (regular) equidistant cylindrical is equal to 1 at the equator! You say it actually yourself at L299, contradicting your response, with the former saying that there are no distortions at the pseudoequator. Snyder (4-15) says "s = h * k * sin(theta). As you have an orthogonal graticule, you have at the equator: 1 * 1 == 1.00 != 1.62.

An equation for areal distortion is missing in the manuscript. As the values for angular and the aspect distortion appear reasonably, please check the areal distortion, or present more clearly and stringent the definition of the areal distortion (or the "normalisation"). Overall, the paper may profit from a better organised section that describes the distortion experiments.

In L481 about the GOF measure, I believe it must be rectified to "bmin is the (global) minimum value for the minor semi-axis across the entire grid plane/partition.", and by this, the GOF is affected by the extent/slicing of grid partitions.

Furthermore, it may be apply for the given DOEC (with b constant and equal 1), but it is not mathematically correct to let GOF reduce to a/b, whereas it should be a*b (when bmin==1)!

2) minor

Give details/show the overlap in Figure 2, and how you obtained the non-overlapping shapes shown in Figure 4. Please comment more (e.g. in the conclusions) that the most unfavourable areas within the (rotated) DOEC are close to the partitions, with highest distortions e.g. over UK or Indonesia.

3) minor

You need also to mention a typical problem of optimised projections: Unusual rotations pose a challenge to laymen and non-professionals, as they need to orientate and cannot plot their country in their well-known shape.

4) minor

L428: I don't understand what is meant by this statement on the water surface (dynamic, procedurally, ...)

-
